# Numerical Study on the Evolution Mechanism of the Crater under a Millisecond Laser

**Dongpo Zhu [1,\*], Peiyun Zhang [1,\*], Zhixiang Tian [1], Cheng Chen [1], Xijun Hua [1], Sheng Xu [2] and Xuan Xie [2]**

[1]  School of Mechanical Engineering, Jiangsu University, Zhenjiang 212000, China; tianzhixiang1996@126.com (Z.T.); JSYZChenC@126.com (C.C.); xjhua@ujs.edu.cn (X.H.)

[2]  School of Automotive and Traffic Engineering, Jiangsu University, Zhenjiang 212000, China; xushengujs@126.com (S.X.); xiexuanujs@163.com (X.X.)

[\*]  Correspondence: zhudongpo1996@163.com (D.Z.); pyzh@ujs.edu.cn (P.Z.)

**Abstract:** A two-dimensional numerical model considering recoil pressure and Hertz-Knudsen ablation rate was established on the foundation of the laser remelting model to investigate the influence of laser processing parameters on crater feature and melted zone, and it was verified through experiments. The temperature and flow velocity distribution of the molten pool during the formation of the crater were analyzed. The results showed that the ablation velocity could be considered under a higher laser peak power density or higher pulse width due to the metal evaporation caused by heat accumulation. The depth and diameter of the crater were significantly affected by laser peak power density and laser pulse duration. Simultaneously, the height of the edge bulge decreased with the increase in pulse duration after 1.5 ms, and the growth rate of central depth was more rapid than that of edge bulge height with the increase of laser peak power density. In the texture with the same depth, a larger melted zone could be obtained with a longer laser duration than the higher peak power density.

**Keywords:** crater morphology; ablation rate; numerical simulation; melted zone; millisecond laser

## 1. Introduction

Laser texturing is a laser processing technology, which can produce a specific micromorphology on the surface of mechanical parts. Because of its advantages like a high processing speed, high efficiency, non-contact processing, and less surface damage, it has received much attention in recent years. The crater texture has both shapes-bulges and dimples where the melted bulges increase the contact strength and friction, while the polished dimples store lubricating oil to form hydrodynamic lubrication [1]. Hence, the crater-shape topography finds wide applications in industry, such as for rollers, piston rings, and bearings [2,3]. Fu et al. [4] studied the effect of the geometrical parameters of the donut-shaped bump on the hydrodynamic lubrication performance. However, unlike for mechanical processing, it was found to be challenging to control the microtexture forming process. Thus, it is important to research the laser melting mechanism and analyze the influence of laser processing parameters on microstructure formation, to realize a self-designed microstructure.

A crater shaped texture can be created by using a millisecond laser. Due to a low peak power density and long pulse duration, the metal substrate mainly melts and solidifies again while interacting with the metal. Hence, many scholars have studied the influence of the Marangoni effect and surface tension on molten pool flow. Based on the equations of conservation of mass, momentum, and energy in the weld pool, He et al. [5] studied the evolution of temperature and velocity fields during laser spot welding of 304 stainless steel by using a heat transfer and fluid flow model, and found that

the liquid flow is mainly driven by the surface tension and to a much lesser extent, by buoyancy force. Traidia et al. [6] took gravity, electromagnetic forces, arc pressure, and the Marangoni effect into consideration and studied the heat transfer and fluid flow in the molten pool. Also, the research compared the influence of various pulsed welding parameters such as pulse frequency and current ratio on the weld quality. Ding et al. [7] studied the Marangoni effect in the model of selective laser melting, and considered that it was the main driving force of the fluid flow and enhanced the heat transfer in the molten pool. In the physical model of Dai et al. [8], while considering the fluid flow driven by the surface tension gradient and influenced by the O element, the terminated distribution state of the AlN reinforcing particle was simulated. Their results showed that the thermo-capillary convection pattern changes from an inward flow pattern to an outward one due to oxidation of the molten pool. Liu et al. [9] established a two-dimensional computational model and a Volume-of-Fluid model to analyze the streamlines and velocity distribution during the laser treatment. Shen et al. [10] established a two-dimensional axisymmetric finite element model to find the contribution of thermo-capillary stress and surface tension to the melt flow and the deformation of the corresponding free surface. The model shows that the Marangoni effect is dominant in the heating phase, while the normal stress is dominant in the cooling period due to the curvature of the free surface.

However, these numerical simulations did not consider the influence of metal evaporation on the molten pool flow after the local temperature reaches the boiling point. It is found that even the millisecond laser can easily make the metal reach its boiling point and that metal evaporation has a significant influence on the melting process [11]. Zhou et al. [12] studied the formation mechanism of the microstructure from the bump to the dimple, and the crater morphology formation was attributed to the intense Marangoni shear stress and recoil pressure. Moreover, when the laser energy is high, recoil pressure is the major driving force. Yuan et al. [13] studied the influence of recoil pressure and Marangoni effect on the ablation morphology of aluminum alloy irradiated with the millisecond-nanosecond combined-pulse laser. Sharama et al. [14] studied the effect of recoil pressure on the surface protuberance height and the genesis of ripple-like structures, and also characterized the surface morphology of the molten pool through the deformation geometry method in the numerical simulation. However, the removal of material during evaporation was not mentioned.

Shen et al. [15] and Ganesh et al. [16] studied the removal of material when the evaporation occurs during laser drilling by using numerical simulations and achieved the separation of evaporated and molten metal by tracking the free surface through the VOF method. Zhang et al. [17] considered the recoil pressure as well as mass loss caused by material evaporation in the level-set model. However, as compared to the deformation geometry method, the level-set method is less effective for the expression of the surface morphology of the texture.

Thus, recoil pressure and ablation velocity are considered in this paper based on the laser remelting deformation geometry model so that the model can be applied more accurately to the analysis of the crater morphology. The influence of laser peak power density and laser pulse duration on the form of crater topography and the melted zone is analyzed according to this model.

## 2. Numerical Simulation

A two-dimensional model with a height of 480 μm and a width of 1500 μm was considered here. The laser source is a Gaussian beam with a diameter of 400 μm and was applied to the center of the upper surface of the model. The material is GCr15, and its physical properties were obtained from JMatPro, as shown in Table 1.

**Table 1.** Chemical composition of GCr15.

| Elements | C | Si | Mn | Cr | Mo | N | Pi | S | Fe |
|---|---|---|---|---|---|---|---|---|---|
| wt% | 1 | 0.2 | 0.34 | 1.5 | 0.01 | 0.08 | 0.013 | 0.003 | Balanced |

Laser irradiation causes an increase in the temperature of the surface of the material, and a molten pool is formed as the temperature exceeds the melting point. The surface of the molten pool was subjected to thermo-capillary stress and Young-Laplace stress, while the interior of the molten pool was subjected to gravity and buoyancy. With the temperature, which continues to rise to the evaporation temperature, a recoil pressure was produced by the evaporated material, which acts on the surface of the molten pool. After the termination of laser irradiation, the molten pool cools and solidifies, the theoretical model of which is shown in Figure 1. Thus, the complete process involved multiple physical processes of heat transfer and fluid flow. Two modules of fluid heat transfer and laminar flow were used to establish the numerical model, based on the following assumptions:

1. The physical properties of materials, obtained from JMatPro, are curves about temperature. For the convenience of calculations, the physical properties of each phase were replaced by constants, as shown in Table 2.

2. The intensity distribution of the laser is considered to follow the ideal Gaussian distribution, and the energy of the process is constant.

3. The solid metal is regarded as a fluid with high viscosity ($5 \times 10^5$ kg/(m·s)). The molten metal in the molten pool was treated as an incompressible Newtonian fluid, while the flow in the molten pool was treated as laminar flow [10].

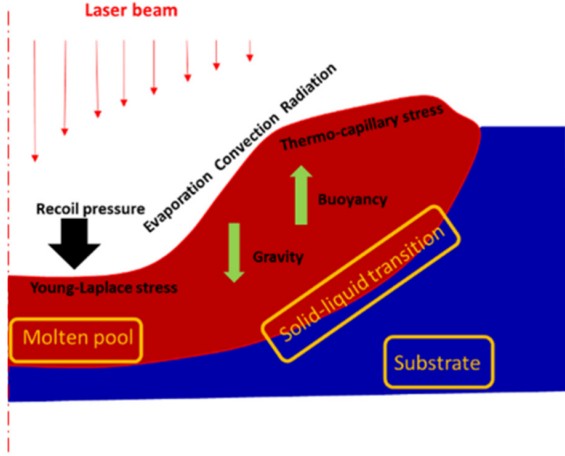

**Figure 1.** Model schematic.

**Table 2.** Material properties of GCr15.

| Property | Symbol | Value |
|---|---|---|
| Liquid phase density | $\rho_l$ | 6500 kg/m$^3$ |
| Solid phase density | $\rho_s$ | 7600 kg/m$^3$ |
| Liquid phase thermal conductivity | $k_l$ | 20 W/(m·K) |
| Solid phase thermal conductivity | $k_s$ | 26 W/(m·K) |
| Melting temperature | $T_m$ | 1700 K |
| Vaporizing temperature | $T_V$ | 2610 K |
| Latent heat of fusion | $L_m$ | $2.2 \times 10^5$ J/kg |
| Latent heat of vaporization | $L_V$ | $6.24 \times 10^6$ J/kg |
| Specific heat of liquid phase | $C_{P\_s}$ | 660 J/kg |
| Specific heat of solid phase | $C_{P\_l}$ | 840 J/kg |
| Surface tension of pure metal | $\gamma_m$ | 1.943 N/m |
| Constant in surface tension gradient | $A_\gamma$ | $3 \times 10^{-4}$ N/(m * K) |
| Mushy zone constant | $A_{mush}$ | $10^7$ |
| Coefficient of heat transfer | $h$ | 10 W/(m$^2$·K) |
| Radiation emissivity | E | 0.7 |
| Retro diffusion coefficient | $\beta_r$ | 0.2 |
| Temperature transition interval of melting | ΔT | 50 K |
| Absorption coefficient | H | 0.39 |

### 2.1. Heat Transfer Governing Equation and Boundary Conditions

#### 2.1.1. Heat Transfer Governing Equation

The transient temperature field was controlled by the energy conservation equation and is expressed as

$$\rho C_P^{eq} \frac{\partial T}{\partial t} + \rho C_P^{eq} u \nabla T = \nabla \cdot (k \nabla T) \tag{1}$$

where $\rho$ is the density, $T$ is the absolute temperature, $t$ is the time, and $k$ is the thermal conductivity.

The latent heat of solid/liquid phase transformation was treated by the equivalent heat capacity method in this paper, and it was simple and convenient to treat the latent heat with this method without distinguishing the solid-liquid phase [18]. $C_P^{eq}$ is the equivalent specific heat, given by:

$$C_P^{eq} = C_P + \frac{L_m}{T_m} H[(T - T_m), \Delta T] + \delta_m L_m \tag{2}$$

$$\delta_m = \frac{exp[-\frac{(T-T_m)^2}{\Delta T^2}]}{\Delta T \sqrt{\pi}} \tag{3}$$

where $L_m$ is the latent heat of melting, $T_m$ is the melting temperature, $C_P$ is the Specific heat capacity at atmospheric pressure, $H[(T - T_m), \Delta T]$ is the Heaviside smoothing function, $\delta_m$ is the normalized Gaussian function near the melting point where the center of the function is the melting point of the target, and $\Delta T$ is the temperature range of the transition between the two phases.

In the simulation, the entire calculation model was regarded as the same phase. To distinguish between the solid, liquid, and the mixed paste area, the volume fraction $f_L$ was used to construct the functions of viscosity $\mu$, thermophysical parameters $k$, $\rho$, and $C_P$.

$$\mu = [1 + (1 - f_L)A_{mush}]\mu_l \tag{4}$$

$$\begin{cases} k = f_L k_l + (1 - f_L)k_s \\ \rho = f_L \rho_l + (1 - f_L)\rho_s \\ C_p = f_L C_{p\_l} + (1 - f_L)C_{p\_s} \end{cases} \tag{5}$$

In the formula above, $\mu_l$ is the dynamic viscosity of liquid phase, $A_{mush}$ is the mushy zone constant, $k_s$, $k_l$, $\rho_s$, $\rho_l$, $C_{P\_s}$, and $C_{P\_l}$ are the thermal conductivity, density, and specific heat capacity at the atmospheric pressure of the solid phase and liquid phase of the material, respectively. The volume fraction $f_L$ changes linearly with temperature in the mushy zone represented as

$$f_L = \begin{cases} 1 & , T > T_l \\ \frac{T - T_s}{T_l - T_s}, & T_s \le T \le T_l \\ 0 & , T < T_s \end{cases} \tag{6}$$

where, $T_s$ is the solid phase temperature and $T_l$ is the liquid phase temperature.

#### 2.1.2. Heat Transfer Boundary Conditions

Herein, it is assumed that the Neumann boundary conditions (natural convection heat transfer, boundary radiation) exist on all surfaces of the target, which are represented as:

$$-k \nabla T = h(T - T_a) + \varepsilon \sigma \left(T^4 - T_a^4\right). \tag{7}$$

Also, the heat source and heat loss due to evaporation and radiation exist on the top surface, shown as:

$$-k \nabla T = Q + h(T - T_a) + \varepsilon \sigma \left(T^4 - T_a^4\right) + Q_V \tag{8}$$

where $h$ is the convection coefficient between the target and the air, $\varepsilon$ is the radiation emissivity, $\sigma$ is the Stefan-Boltzmann constant, $T_a$ is the ambient temperature, $Q_V$ is the evaporation heat loss, and the specific expression is given by

$$Q_V = M_V \times L_V \tag{9}$$

here, $M_V$ is the vaporized mass flow rate and $L_V$ is the latent heat of vaporization.

$$M_V = \sqrt{\frac{m}{2\pi R T_s}} \times P_{sat}(T) \times (1 - \beta_r) \tag{10}$$

$$P_{sat}(T) = P_{atm} \times exp\left(\frac{M_a L_v}{R}\left(\frac{1}{T_v} - \frac{1}{T}\right)\right) \tag{11}$$

In this formula, $m$ represents the atomic mass, $P_{sat}(T)$ represents the saturated vapor pressure, $\beta_r$ is the retro diffusion coefficient, $R$ is the Boltzmann constant, $T_v$ is the vaporization point of the material, $L_v$ is the latent heat of vaporization, and $P_{atm}$ is the standard atmospheric pressure [19].

Also, $Q$ is the laser heat source, and a Gaussian surface heat source is used. Its specific expression is as follows:

$$Q = 2\eta I_{laser} e^{-\frac{2x^2}{r^2}} \tag{12}$$

where $r$ is the effective spot radius of the pulse laser, $I_{laser}$ is the laser peak power density, $\eta$ is the absorption coefficient, and $x$ is the distance between the transverse direction and the focus.

### 2.2. Laminar Governing Equations and Boundary Conditions

#### 2.2.1. Laminar Governing Equations

In the following heat transfer equations, $u$ is the velocity field of the molten pool in the momentum conservation formula (Navier-Stokes equation), given by:

$$\rho\frac{\partial u}{\partial t} + \rho u \cdot \nabla u = \nabla\left(\mu(\nabla u + (\nabla u)^2 - pI)\right) + F_v \tag{13}$$

$$F_v = F_g + F_b = \rho g - \beta(T - T_m)\rho g \tag{14}$$

where $F_v$ is the volume force, $F_g$ is the gravity, $F_b$ is the buoyancy, and $g$ is the acceleration of gravity.

#### 2.2.2. Laminar Boundary Conditions

The upper surface is an open boundary, which can be deformed freely as Young-Laplace stress and recoil pressure act in the normal direction. The thermo-capillary stress acts on the tangent direction of the surface, represented as follows [17]:

$$\sigma_n = -P_{recoil} \cdot n + 2k\gamma \cdot n \tag{15}$$

$$\sigma_t = \frac{\partial \gamma}{\partial T}\nabla_s T \cdot t \tag{16}$$

$$\gamma = \gamma_m - A_\gamma(T - T_m) \tag{17}$$

where $k = -\nabla \cdot n$, $k$ is the curvature of the surface profile, $\gamma$ is the surface tension, $\frac{\partial \gamma}{\partial T}$ is the surface tension coefficient with temperature, $\nabla_s T$ is the temperature gradient along the tangent direction of the surface, $\gamma_m$ is the surface tension of the pure metal at the melting point, $A_\gamma$ is the surface tension temperature coefficient of the pure metal, and $n$ and $t$ are the unit normal vector and unit tangent vector of the surface, respectively.

The vapor recoil pressure, generated by evaporation at the gas-liquid interface after the surface of the molten pool exceeds the vaporization point, is [20]

$$P_{recoil} = \begin{cases} P_{atm}, 0 \leq T < T_v \\ \frac{1+\beta_r}{2} \times P_{sat}(T_s), T \geq T_v \end{cases}.$$  (18)

The influence of environmental pressure is also considered in this model, in which the surface changes invisibly when the atmospheric pressure is 1 atm.

### 2.3. Deformation Geometry and Meshing Mesh

In this paper, the deformation geometry method and Laplace smoothing method are used to trace the gas-liquid interface. The movement of the boundary nodes is controlled by the velocity of the fluid and ablation rate, which is theoretically based on the Hertz-Knudsen equation, and the expressions are

$$u_{mesh} = u - V_{ex}$$  (19)

$$v_{mesh} = v - V_{ey}$$  (20)

$$V_e = \frac{M_V}{\rho}$$  (21)

where $u_{mesh}$ is the transverse velocity of the grid, $v_{mesh}$ is the longitudinal velocity of the grid, $u$ is the transverse velocity of the fluid, $v$ is the longitudinal velocity of the fluid, $V_{ex}$ is the component of the ablation velocity in the $x$ direction, and $V_{ey}$ is the component of the ablation velocity in the $y$ direction.

It is to be noted that the horizontal axis of the two-dimensional model is $x$, and the vertical axis is $y$. In order to improve the calculation accuracy and reduce the calculation cost, the minimum unit size of the upper surface area is 0.07 μm, the minimum unit size of the rest of the areas is 1.44 μm, the unit growth rate is 1.2, the curvature factor is 0.4, the number of units is 9949, and the average mesh quality is 0.876. The mesh formation is shown in Figure 2. MUMPS solver is used in the numerical model where the relative tolerance is 0.001, the tolerance factor is 0.1, and the time step is 0.01 ms.

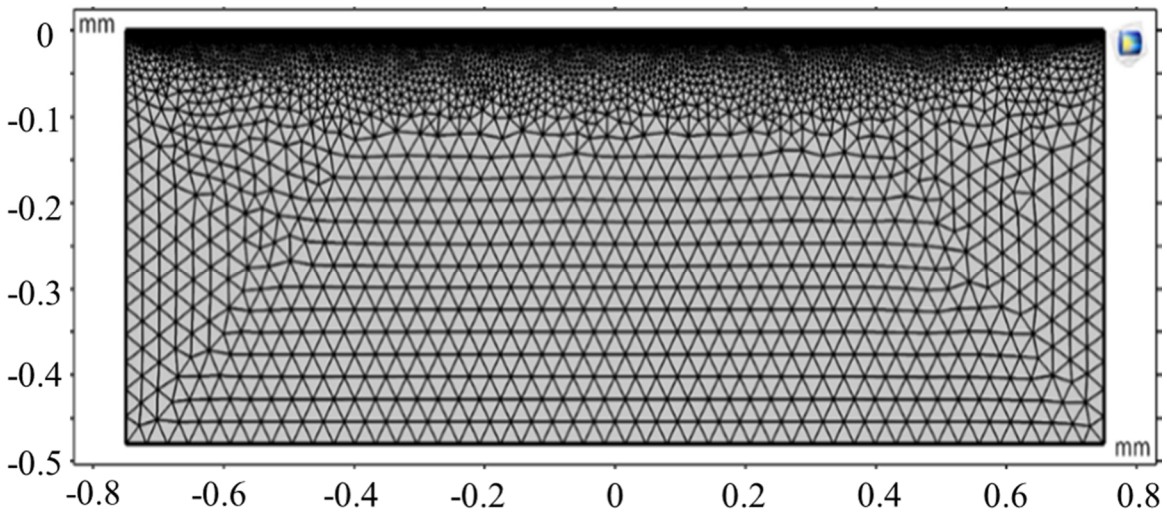

**Figure 2.** Finite element mesh.

### 2.4. Experimental Method

In the verification experiment, GCr15, after polishing, was selected, and its surface roughness reached Ra = 0.05 μm. The samples were processed by a millisecond laser with argon as the auxiliary gas. After processing, the samples were put into an ultrasonic cleaning device for cleaning, and the

surface profile of the samples was then observed and measured by the NanoFocus μsurf explorer confocal microscope system. Each set of parameters was repeated 10 times, the average value of the crater profile was calculated, and the morphology close to the average value was compared with the simulation results. The laser used in the experiment was a self-developed millisecond multi-function machining system with a wavelength of 1064 nm, a focal length of 25 mm, and spot diameter of 400 μm. A schematic diagram of the crater is shown in Figure 3.

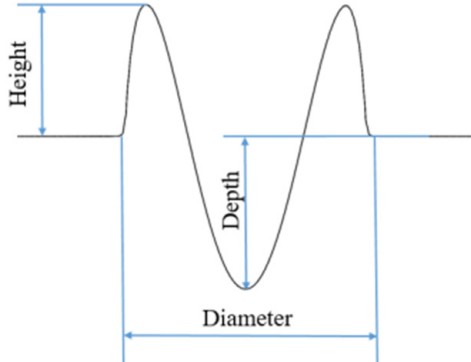

**Figure 3.** The schematic of the crater.

## 3. Results and Discussion

### 3.1. Effect of Ablation Rate on Morphology and Verification of Numerical Model

Figure 4 illustrates the comparison of microstructure morphology after cooling, with and without the ablation amount, when the peak power density is $3 \times 10^9$ W/m$^2$. It can be seen that the two curves have a high coincidence degree with a height of about 1.55 μm, a depth of about 1.78 μm, and a radius of about 110 μm, which are consistent with the experimental results (Figure 5). This can be considered from the temperature distribution diagram (Figure 6), where the highest temperature at 1 ms is only 2650 K and the molten pool flows outwardly at a maximum velocity of 0.21 m/s. Although ablation has already taken place at this time, its corresponding ablation speed is lower (as shown in Figure 7), which is derived from Equation (21). In this case, the higher the temperature is, the more drastic the ablation speed would be. Thus, the ablation speed corresponding to 2650 K is around to 0.5 mm/s. Consequently, the influence of the ablation amount at the peak power density could be ignored.

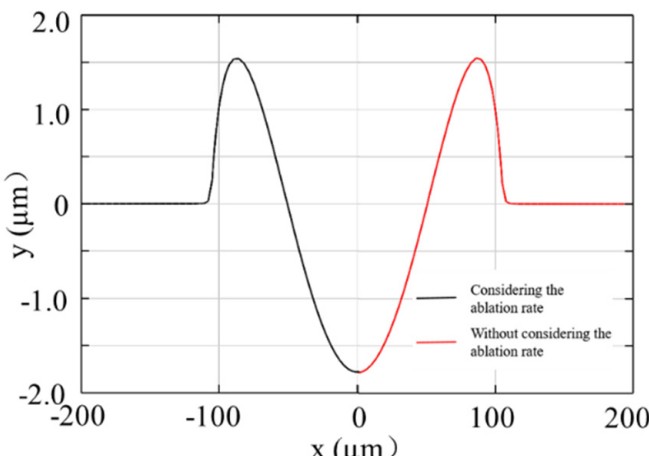

**Figure 4.** Comparison of molten pool with and without the ablation rate for $I = 3 \times 10^9$ W/m$^2$, $\tau = 1$ ms, $t = 10$ ms.

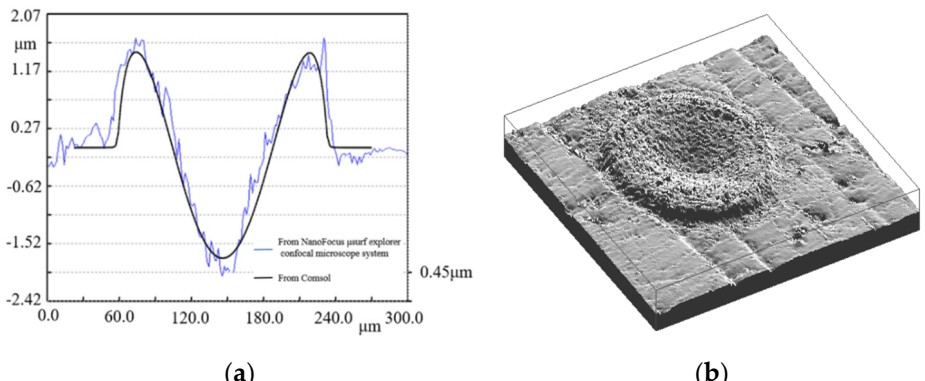

(**a**)　　　　　　　　　　　　　　　　　　(**b**)

**Figure 5.** (**a**) Comparison of crater morphology from Comsol and NanoFocus µsurf explorer confocal microscope system for $\tau = 1$ ms, $I = 3 \times 10^9$ W/m$^2$, (**b**) Three-dimensional diagram of a crater from the NanoFocus µsurf explorer confocal microscope system.

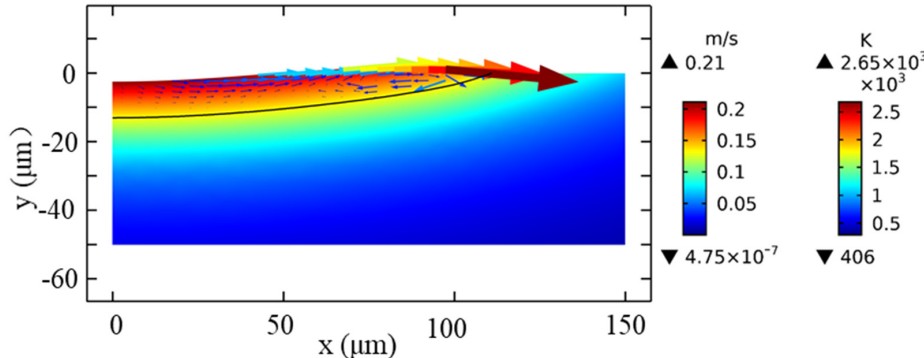

**Figure 6.** Transient melt flow during laser heating for $t = 1$ ms, $\tau = 1$ ms, $I = 3 \times 10^9$ W/m$^2$.

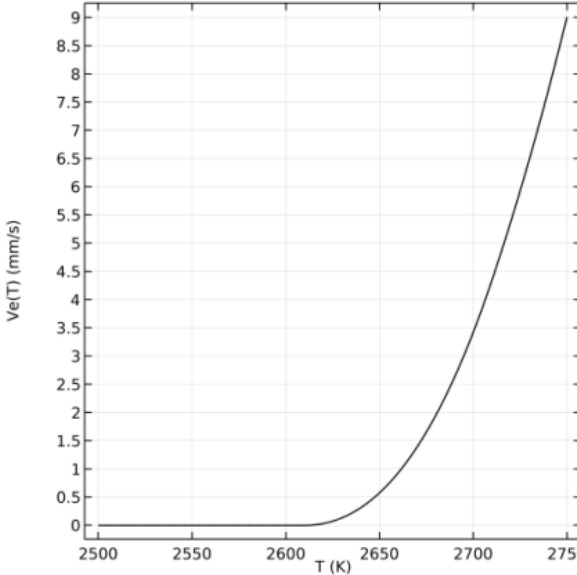

**Figure 7.** Variation curve of the ablation rate with the temperature from Equation (21).

Figure 8 compares the surface morphology of molten pool with and without the Hertz-Knudsen ablation rate, when the peak power density is $5 \times 10^9$ W/m$^2$ and the pulse width is 1 ms. It can be seen that the difference between the diameters and depths of the molten pool surface is less than 0.5 µm, while the difference between the heights is more than 3.5 µm at the end of laser heating. The temperature and flow velocity distribution of the molten pool, at 1 ms, are, respectively, shown in

Figure 9. The maximum temperature is 2750 K, while the corresponding ablation velocity is around 9 mm/s, which results in the reduction of the molten pool and its flow to the edge. In addition, the removal of materials reduces the temperature from 2750 K to 2740 K, which weakens the influence of recoil pressure and causes the flow rate to decrease from 0.64 m/s to 0.46 m/s; it also reduces the accumulation at the edge of the molten pool.

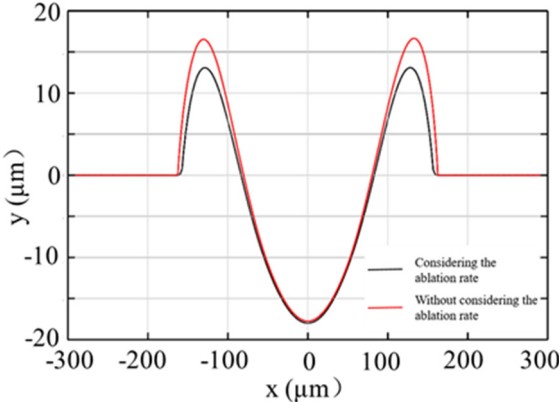

**Figure 8.** Comparison of the molten pool with and without the ablation rate for $I = 5 \times 10^9$ W/m$^2$, $\tau = 1$ ms, $t = 1$ ms.

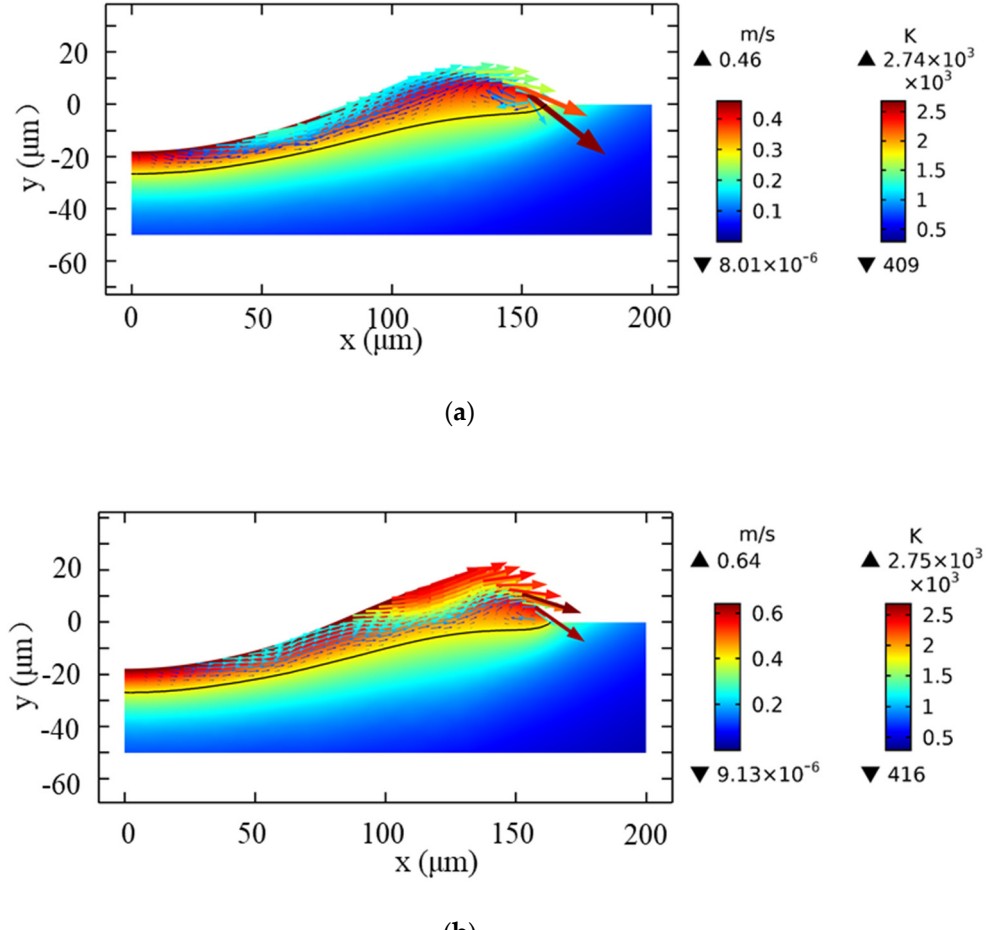

**Figure 9.** Transient melt flow during laser heating for $t = 1$ ms, $\tau = 1$ ms, $I = 5 \times 10^9$ W/m$^2$, temperature field (color surface contour), and velocity field (colored arrow plots). (**a**) Ablation rate considered (**b**) Without the ablation rate.

Figure 10 shows the temperature and flow velocity distribution diagrams in the cooling stage of 1.03 ms. It can be seen that the melt in the molten pool has a flow velocity of 1.39 m/s (when the ablation rate is ignored) and 1.14 m/s (when the ablation rate is considered). It can be considered that the height of the edge bulge possesses a positive correlation with the curvature of the molten pool surface, which makes the Young-Laplace stress increase. Consequently, the molten pool was affected by it and possessed a higher flow rate and reflux quantity in the cooling stage, making the center depth of the microstructure decrease, compared to the addition due to the Hertz-Knudsen ablation rate. The microstructure morphology, formed after cooling, is shown in Figure 11. It can be seen that the edge of the microstructure had a higher bulge and shallower depth when the ablation amount was ignored. After adding the ablation speed, the edge bulge became smaller with deeper depth, which was in favorable consistency with the experimental results shown in Figure 12, and verified the simulation's correctness.

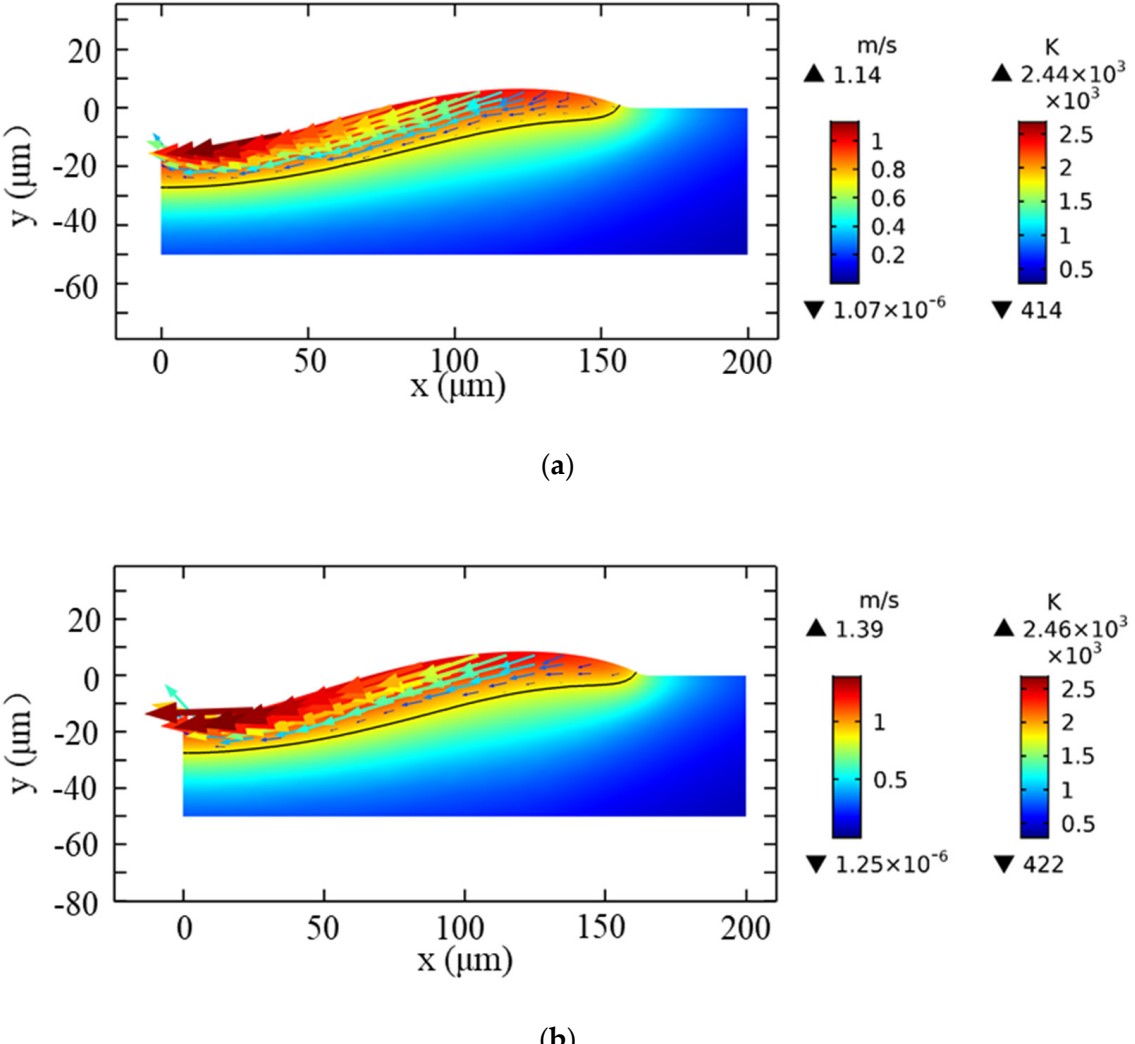

(a)

(b)

**Figure 10.** Transient melt flow for $t = 1.03$ ms, $\tau = 1$ ms, $I = 5 \times 10^9$ W/m$^2$, temperature field (color surface contour), and velocity field (colored arrow plots). (**a**) The ablation rate was considered (**b**) Without the ablation rate.

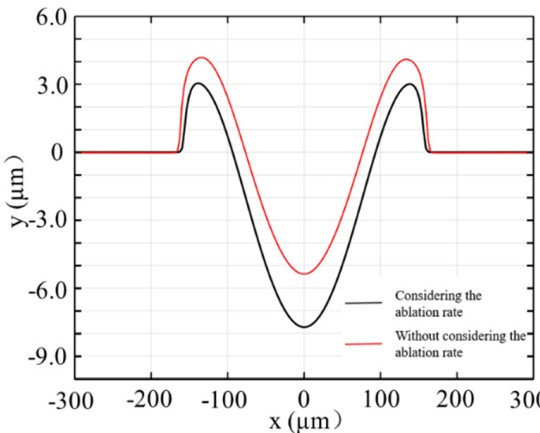

**Figure 11.** Comparison of the two-dimensional morphology with and without the ablation rate for $t = 10$ ms, $\tau = 1$ ms, $I = 5 \times 10^9$ W/m$^2$.

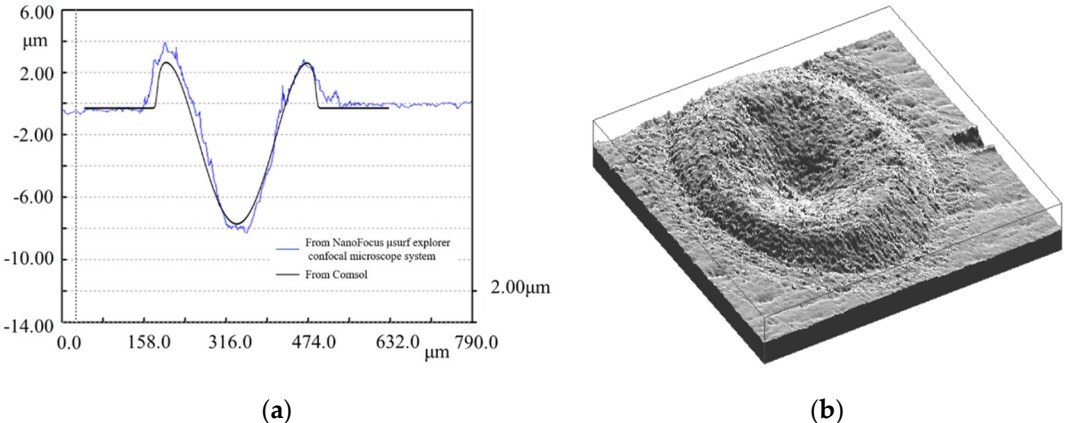

(**a**)                                                    (**b**)

**Figure 12.** (**a**) Comparison of a crater morphology from the Comsol and NanoFocus µsurf explorer confocal microscope system for $\tau = 1$ ms, $I = 5 \times 10^9$ W/m$^2$, (**b**) Three-dimensional diagram of a crater from the NanoFocus µsurf explorer confocal microscope system.

The crater morphology simulated with and without ablation rate changes with the increase in pulse width is shown in Figure 13. Hence, the ablation rate should not be ignored in the study of crater morphology formation.

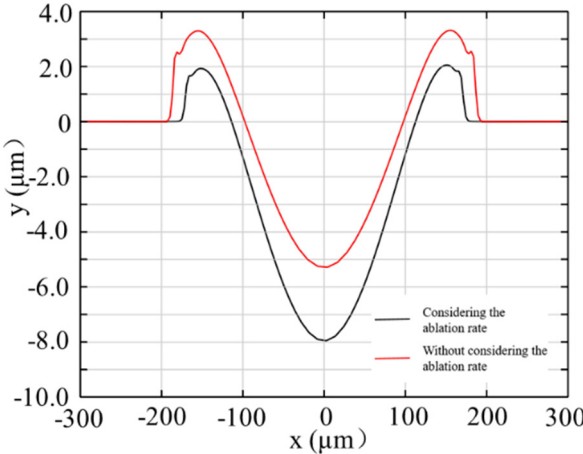

**Figure 13.** Comparison of molten pool with and without the ablation rate for $I = 3 \times 10^9$ W/m$^2$, $\tau = 3$ ms, $t = 10$ ms.

### 3.2. Effect of Peak Power Density on Morphology

　　The microstructure morphology and melted zone under the peak power density of $3 \times 10^9 - 5.5 \times 10^9$ W/m$^2$ were compared, and the melted zone area was calculated by integrating the closed area composed of 1700 K isotherm and the surface curve (as shown in Figure 14) at the end of laser heating. It can be seen from Figure 15 that the experimental and simulation results show the same variation trend, and the difference in the morphology may be attributed to the differences between the thermophysical parameters of the material and laser parameters in the numerical simulation and the experiment. The simulation results also show that with an increase in the laser peak power density, the height of the edge bulge increases from 1.5 μm to 3.4 μm, the center depth increases from 1.78 μm to 9.95 μm, the diameter increases from 110 μm to 170 μm, and the melted zone increases from 0.00198 mm$^2$ to 0.00648 mm$^2$. On the other hand, the growth rate of central depth is more rapid than that of the edge bulge height.

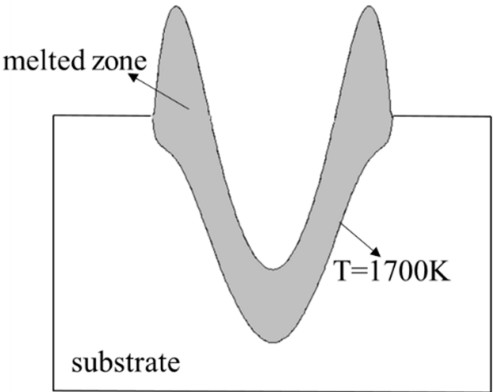

**Figure 14.** Integral diagram of the melted zone area.

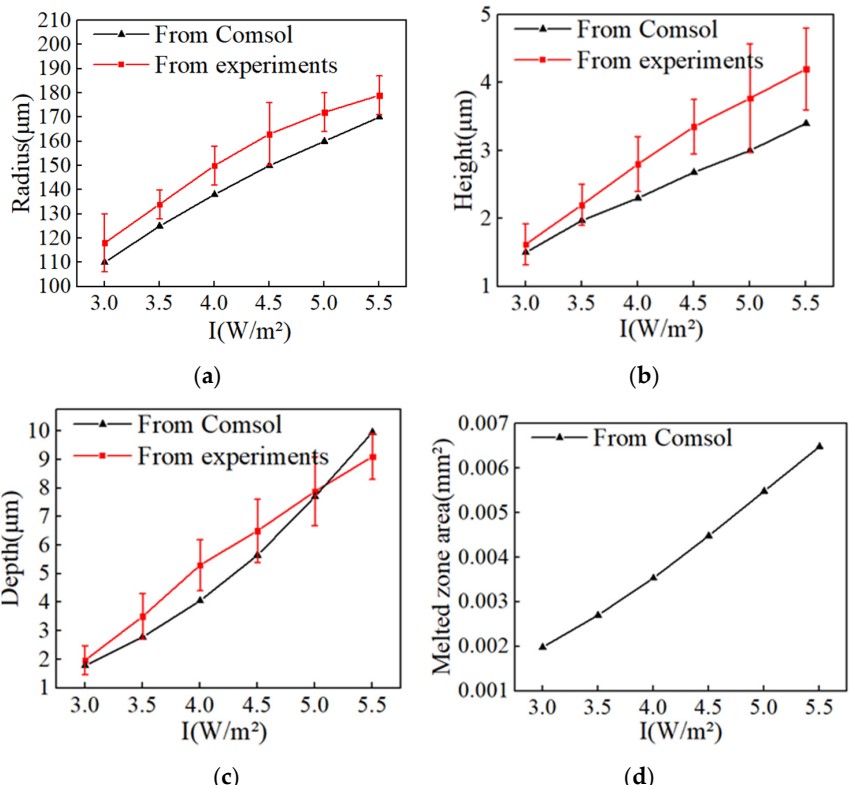

**Figure 15.** Variation in the molten pool under different peak power densities for τ = 1 ms, t = 10 ms. (**a**) Radius of the crater, (**b**) Height of the crater, (**c**) Depth of the crater, (**d**) Melted zone area of the crater.

At the end of the heating process, the surface morphology of the molten pool was compared, as shown in Figure 16. It can be seen that the growth rate of the molten pool depth was also more rapid than that of the edge bulge height. This can be attributed to the fact that the bulge at the edge was formed by the outward flow and accumulation of molten metal in the molten pool, under the influence of recoil pressure and the thermo-capillary stress, and the amount of molten material accumulated at the edge was reduced due to the existence of ablation rate. The increase of center depth was finally more evident than edge bulge height after cooling.

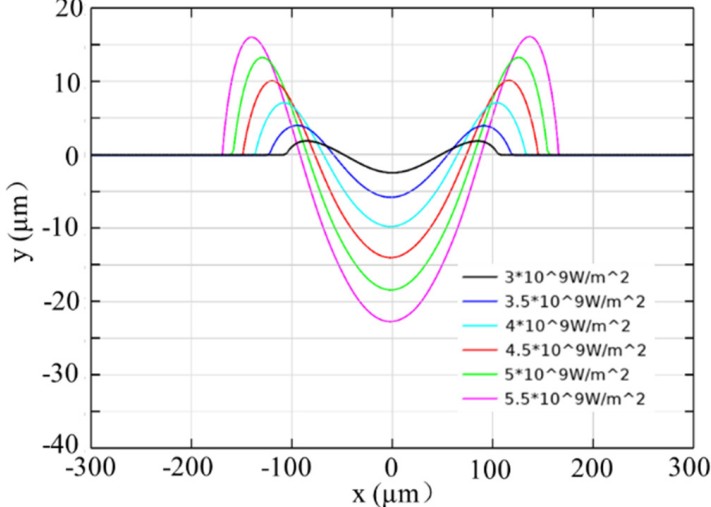

**Figure 16.** Crater morphology under different peak power density for $t = 10$ ms, $\tau = 1$ ms.

### 3.3. Effect of Laser Duration on Morphology

The effects of the ablation rate on microstructure morphology were compared when the power density was $3 \times 10^9$ W/m$^2$ and the laser duration was 1–3.5 ms. Similarly, the experimental and simulated results showed the same variation trend. It can be seen from Figure 17 that the depth, diameter, and melted zone of the microstructure increased with the increase of the laser pulse width and as the depth increased from 1.78 μm to 9.45 μm, the radius increased from 110 μm to 186 μm and the melted zone increased from 0.00198 mm$^2$ to 0.01249 mm$^2$, while the height of the edge bulge, in the linear diagram of Figure 17b, is shown to decrease with the increase of pulse width after 1.5 ms.

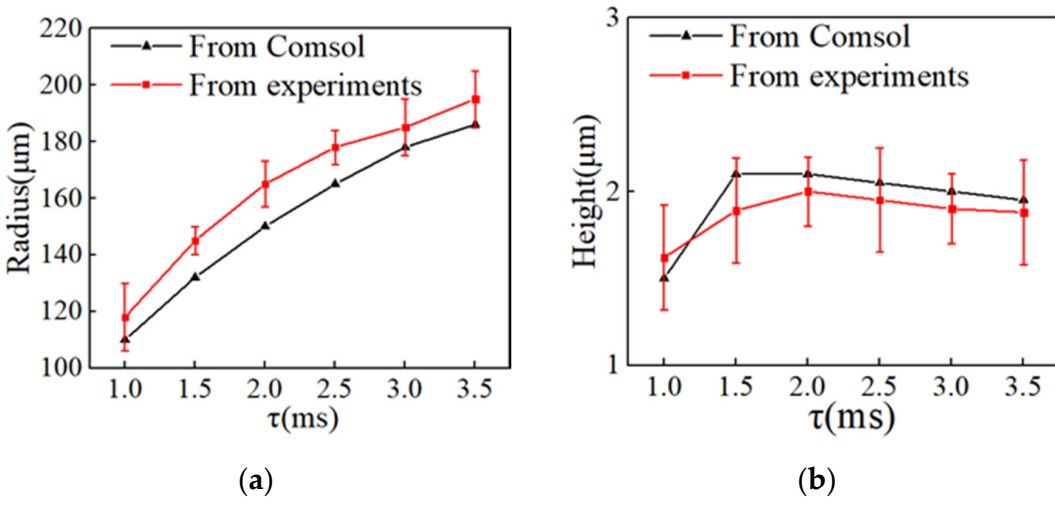

(a)　　　　　　　　　　　　　　　　　　　　(b)

**Figure 17.** *Cont.*

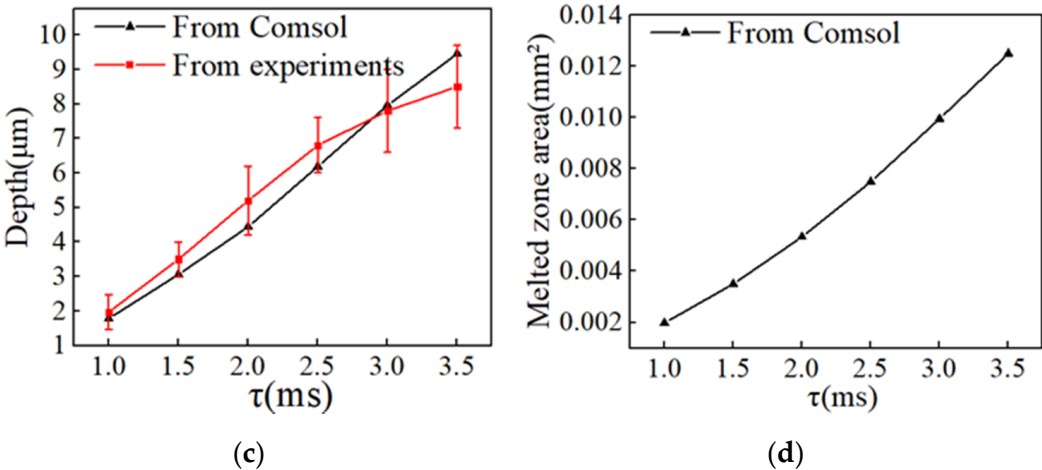

**Figure 17.** Variation of the molten pool under different laser durations for $I = 3 \times 10^9$ W/m$^2$, $t = 10$ ms. (**a**) Radius of the crater, (**b**) Height of the crater, (**c**) Depth of the crater, (**d**) Melted zone area of the crater.

The surface morphology of the molten pool at the end of heating under a different pulse width was compared, as shown in Figure 18. It should be noted that when the pulse width changes, the central depth of the molten pool changes more sharply, and the height of the edge bulge increases more slowly in comparison. The reason for this is that the maximum temperature of the molten pool and the evaporation time of materials increases with the increase of laser duration, leading to more material being ablated and removed. Due to the clear difference between the pool depth and bulge height, more molten reflow is performed during cooling under the effect of Young-Laplace stress, resulting in a decrease of the edge bulge after cooling.

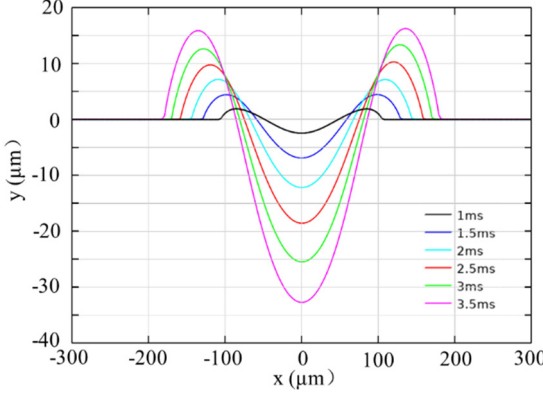

**Figure 18.** Crater morphology under a different laser duration for $t = 10$ ms, $I = 3 \times 10^9$ W/m$^2$.

In engineering applications of lubrication and friction, a deep crater is usually required, and a larger melted zone is beneficial for improving the strength of the contact surface. Upon comparing the microstructures with a peak power density of $5 \times 10^9$ W/m$^2$, a pulse width of 1 ms, a peak power density of $3 \times 10^9$ W/m$^2$, and pulse width of 3 ms, as shown in Figures 15 and 17, it can be found that the two microstructures have similar depths after cooling. The surface morphology of the molten pool at the end of heating is presented in Figure 19. It can be seen that the depths of the two microstructures are quite different at 18 μm in Figure 19a and 25.2 μm in Figure 19b, whereas the height of the edge bulge is similar, at 13 μm in Figure 19a and 12.5 μm in Figure 19b. The edge bulge width of the latter is more extensive, which makes more melt flow back, resulting in similar depths after cooling. Moreover, it can be seen that the melting area is larger under the high pulse width. Therefore, it can be concluded that when rhe laser pulse width is between 1 ms and 3.5 ms, peak power density is between

$3 \times 10^9$ W/m$^2$ and $5 \times 10^9$ W/m$^2$, z higher pulse width will lead to a larger melted zone than a higher power density upon obtaining microstructures with the same depth.

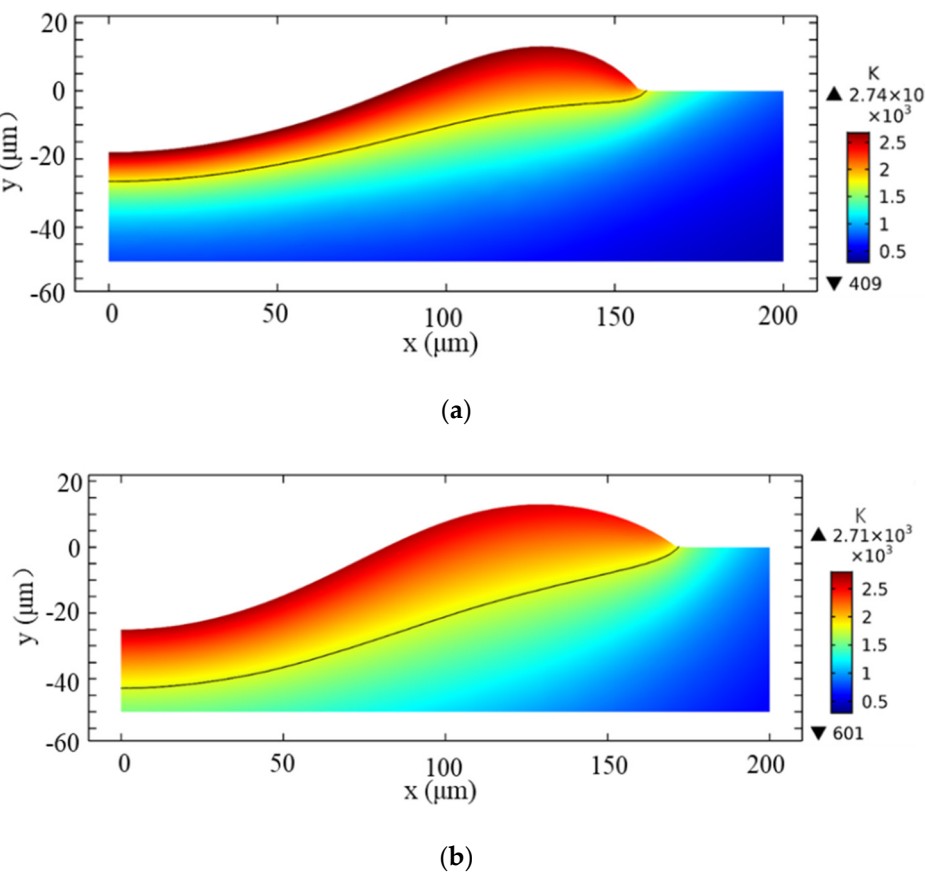

**Figure 19.** Transient temperature field (color surface contour) for $t = 1.03$ ms, $\tau = 1$ ms, $I = 5 \times 10^9$ W/m$^2$. (**a**) $I = 5 \times 10^9$ W/m$^2$, $\tau = 1$ ms, $t = 1$ ms (**b**) $I = 3 \times 10^9$ W/m$^2$, $\tau = 3$ ms, $t = 3$ ms.

## 4. Conclusions

In this paper, a two-dimensional numerical model for studying the crater morphology and its melted zone was established. In addition to the Marangoni effect, surface tension and volume force, recoil pressure, and the Hertz-Knudsen ablation rate were also considered. The model was verified experimentally, and the significant conclusions of this research are summarized as follows:

The effect of ablation velocity on the crater morphology can be ignored at a lower peak power density ($<3 \times 10^9$ W/m$^2$) and lower laser duration ($<1$ ms). However, when the peak power density increases or the laser duration increases, the effect of the ablation rate on the crater formation becomes more evident.

With the increase of peak power density and laser duration, the material evaporates and the molten pool expands because of heat accumulation. This makes the diameter and the center depth of the crater texture increase after the cooling period. However, with the increase of laser duration, the height of the edge bulge decreases, which is due to the fact that the central depth of the molten pool morphology is too large in comparison with the edge bulge at the end of heating, which requires more molten reflow during the cooling period.

The laser peak power density and pulse width obviously affect the melted zone. A numerical model is used to explain that in a crater with the same depth, a larger melted zone can be obtained with a higher laser duration time than with higher peak power density, and this model is effective for studying the formation of the crater under a millisecond laser.

**Author Contributions:** The contributions of the individual authors were the following: methodology, D.Z. and P.Z.; numerical simulation, D.Z., P.Z., C.C., X.X., and S.X.; experimental validation, D.Z., Z.T., and P.Z.; writing—original draft preparation, D.Z. and P.Z.; writing—review and editing, D.Z., P.Z., and Z.T.; funding acquisition, X.H. All authors have read and agreed to the published version of the manuscript.

**Funding:** This research was supported by the National Natural Science Foundation of China (Grant No. 51975252), High-technology Research Key Laboratory Project of Zhenjiang City (SS2018007), Major Scientific and Technology Project of Zhenjiang City (ZD2018001), Major Scientific and Technological Achievement Transformation Project of Taizhou City (SCG201901).

**Conflicts of Interest:** The authors declare no conflict of interest.

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
