# Peer review of "Numerical Study on the Evolution Mechanism of the Crater under a Millisecond Laser"

_applsci, doi:10.3390/app10249054_

Round 1
Reviewer 1 Report
The paper simulated crater morphology tests with and without ablation rate changes, the results showed that the model considering ablation rate fits the actual shape of crater better as the ablation rate is negligible at high temperature. In addition, the author studied the relations between various crater morphology (center depth, diameter and melted zone of microstructure) and process parameters (laser pulse width and laser peak density). In short, a more accurate model was established, which considered the recoil pressure and ablation velocity. However, some areas may require clarification or modification.
- The data source of ablation rate with temperature (figure 7) is not mentioned.
- Regarding the crater morphology captured by NanoFocus μsurf explorer confocal microscope system in figure 5 (a), does the profile come from the average of multiple profile sections?
- Some relations in the paper are concluded as “linear”, such as the depth, diameter and melted zone of microstructure with laser pulse width. However, it may be necessary to calculate a linear regression model in order to conclude that two variables are linear.
- The unit of the abscissa in figure 17 (b) should be τ(ms).
- Line 297 : “needed” → “performed”?
- Line 336: The melted zone is directly proportional to the laser peak power density and pulse width.
It may not be appropriate to use "proportion" because there is no proof that the two variables have a constant ratio.
- Line 310-312: Need to add the specified range: Laser pulse width from 1 ms to 3.5 ms, peak power density from 3*109 to 5.5*109 W/m2.
Author Response
Dear Reviewer,
We appreciate for your warm work earnestly.
Special thanks to you for your good comments.
Please see the attachment.
Yours sincerely,
Dongpo Zhu

Reviewer 2 Report
The authors present the two-dimensional numerical model for the ms laser ablation mechanism on solid metal. The proposed manuscript within the scope of the IJAMT journal. Some of the results observations are interesting, however, if the paper can be improved in the following areas, it would add more value to the readers: a) The literature review should be added recent five years paper in the manuscript. b) For the industrial application field should be described. c) Experiments: how many times for each condition evaluation. The standard and bias should be added to the present results. d) The theoretical model part needs to be verified with experiments to confirm the reliability.Author Response
Dear Reviewer,
We appreciate for your warm work earnestly.
Special thanks to you for your good comments.
Please see the attachment.
Yours sincerely,
Dongpo Zhu

Reviewer 3 Report
The paper deals with crater formation during millisecond laser ablation.
Millisecond laser ablation is out of the actual research focus because shorte pulse length deliver far better ablation quality. The achived results are not new in the laser ablation community for example in the conclusion:
"The melted zone is directly proportional to the laser peak power density and pulse width. Considering a crater with the same depth, a larger melted zone can beobtained with higher laser duration time than with higher peak power density."
Such statements are widely known.
In addition similar results have alredy been presented priviously:
"The Effect of Spot Size Combination Mode on Ablation Morphology of Aluminum Alloy byMillisecond-Nanosecond Combined-Pulse Laser": Bo-Shi YuanID, Ye Zhang, Wei Zhang, Yuan Dong and Guang-Yong Jin
Further there are several errors:
Some figures are only described by one sentence for example figure 4 to 7.
Figure 6 and similar figures depict velocity vectors which are very difficult to see.
Figure 15 and 17 have very small and illegible labels
The model shold be better validated by experiments. Not only with 2 figures.
Eperimental results should also inserted in figure 15 and 17
Author Response

(The authors gave the same response as above.)

Round 2
Reviewer 1 Report
The authors have responded to all my questions and made necessary changes to the manuscript
Reviewer 2 Report
Authors have been revised reviewer’s comment.
I this status, I suggestion this manuscript can be accepted for publication. However, the english should be publish by grammar correct.
Reviewer 3 Report
The authors considered the reviewers remarks well.